# A Robust Mean Teacher Framework for Semi-Supervised Cell Detection in Histopathology Images

**Ziqi Wen**          3220210669@BIT.EDU.CN and **Chuyang Ye**          CHUYANG.YE@BIT.EDU.CN
*School of Integrated Circuits and Electronics, Beijing Institute of Technology, Beijing, China*

**Editors:** Accepted for publication at MIDL 2023

## Abstract

Cell detection in histopathology images facilitates clinical diagnosis, and deep learning methods have been applied to the detection problem with substantially improved performance. However, cell detection methods based on deep learning usually require a large number of annotated training samples, which are costly and time-consuming to obtain, and it is desirable to develop methods where detection networks can be adequately trained with only a few annotated training samples. Since unlabeled data is much less expensive to obtain, it is possible to address this problem with semi-supervised learning, where abundant unlabeled data is combined with the limited annotated training samples for network training. In this work, we propose a semi-supervised object detection method for cell detection in histopathology images, which is based on and improves the mean teacher framework. In standard mean teacher, the detection results on unlabeled data given by the teacher model can be noisy, which may negatively impact the learning of the student model. To address this problem, we propose to suppress the noise in the detection results of the teacher model by mixing the unlabeled training images with labeled training images of which the ground truth detection results are available. In addition, we propose to further incorporate a loss term that is robust to noise when the the student model learns from the teacher model. To evaluate the proposed method, experiments were performed on a publicly available dataset for multi-class cell detection, and the experimental results show that our method improves the performance of cell detection in histopathology images in the semi-supervised setting.

**Keywords:** Cell detection, Histopathology image analysis, Semi-supervised learning

## 1. Introduction

Automated cell detection in histopathology images allows quantitative, efficient, and reproducible cell analysis, and it can benefit diagnostics and disease grading in clinical practice (van der Laak et al., 2021). Methods based on deep learning have been developed for the cell detection task and achieved promising results. For example, earlier works have used customized deep networks to identify the locations of cell nuclei (Xu et al., 2016; Sirinukunwattana et al., 2016), and more recent works have used or adapted modern object detectors, such as Faster R-CNN (Ren et al., 2017), for cell detection (Cai et al., 2019; Sun et al., 2021), where bounding boxes are generated to localize the cells of interest.

The training of these cell detection models based on deep learning usually requires a large amount of labeled training data, where the cells of interest need to be annotated on a sufficient number of training images. However, because the cell morphology in histopathology images can be diverse and the annotation should be performed by well-trained experts, the collection of a large-scale annotated dataset for training can be challenging, and often

only a limited amount of labeled training data is available. Therefore, it is desirable to develop cell detection approaches that perform well given scarce labeled training data.

One common strategy to address the problem of scarce labeled training data is to exploit *semi-supervised learning* (SSL) (Cheplygina et al., 2019), where unlabeled data that is easy to obtain and thus in general abundant is used together with the scarce labeled training data during network training. The SSL strategy can be applied to object detection problems as well. For example, in (Sohn et al., 2020) and (Liu et al., 2020), the teacher-student framework is adapted to object detection, where the student model learns from not only labeled training data but also unlabeled training data with predictions given by the teacher model. The teacher model can be fixed (Sohn et al., 2020) or gradually updated during network training (Liu et al., 2020). These SSL-based object detection methods can be readily applied to cell detection. However, due to the limited amount of labeled training data, the predictions given by the teacher model for unlabeled training images can be noisy and prone to errors, which negatively affects the training of the student model and the final detection performance, and the development of SSL-based methods for cell detection in histopathology images is still an open problem.

To further explore the problem of semi-supervised cell detection in histopathology images, we seek to improve the teacher-student interaction by better handling the noisy teacher predictions. Like (Ying et al., 2021), we develop a semi-supervised cell detection method based on the *mean teacher* (MT) framework (Tarvainen and Valpola, 2017). Unlike (Ying et al., 2021), instead of directly using the unlabeled training data and the corresponding noisy teacher predictions to train the student model, we propose to mix the unlabeled images with the labeled images to produce synthetic unlabeled images for network training. Specifically, each region detected in the unlabeled images by the teacher model is mixed with a region from the labeled images belonging to the same cell type of the detected region, and the classification of the mixed region is unchanged. Because the information in the labeled images without label noise is included in the synthetic unlabeled images, compared with the teacher predictions for the original unlabeled images, the label noise in the synthetic unlabeled images is suppressed. These synthetic unlabeled images and their synthetic labels then replace the original unlabeled images and their teacher predictions for training the student model. In addition, to better deal with the noise in the synthetic labels, we propose to incorporate a loss term that is more robust to label noise when the student model learns from the teacher model. Like in the standard MT framework, in our method the teacher model is also updated during network training based on the learned student model with *exponential moving average* (EMA) (Tarvainen and Valpola, 2017), and the final teacher model is used to perform cell detection. To evaluate the proposed method, we performed experiments on a publicly available dataset for cell detection in histopathology images. The experimental results indicate that the proposed method improves the performance of semi-supervised cell detection in histopathology images.

## 2. Methods

### 2.1. Problem Formulation

The aim of this work to improve cell detection in histopathology images in the semi-supervised setting, where only a limited number of labeled training images and a large

number of unlabeled training images are available. We focus on the use of modern object detectors, where the detection result is represented by a bounding box that indicates the location of the detected cell and the probability of the cell belonging to certain types (Ren et al., 2017). For the labeled training images, the annotation gives the ground truth location and type of the cells of interest, whereas for the unlabeled training images the ground truth is unknown.

In the typical fully supervised setting, only the network prediction and ground truth information in the labeled images are used for network training. For convenience, we denote the localization and classification results of the $m$-th detected cell in the labeled images by $x_m^{\mathrm{a}}$ and $c_m^{\mathrm{a}}$, respectively, and the total number of detected cells in the labeled images is denoted by $N_{\mathrm{a}}$. Then, the following loss function that takes the localization and classification error into consideration is minimized to learn the weights of the detection network:

$$\mathcal{L}_{\mathrm{sup}} = \frac{1}{N_{\mathrm{a}}} \sum_{m=1}^{N_{\mathrm{a}}} J(x_m^{\mathrm{a}}, y_m^{\mathrm{a}}) + H(c_m^{\mathrm{a}}, d_m^{\mathrm{a}}), \tag{1}$$

where $J(x_m^{\mathrm{a}}, y_m^{\mathrm{a}})$ measures the difference between $x_m^{\mathrm{a}}$ and the corresponding ground truth location $y_m^{\mathrm{a}}$, and $H(c_m^{\mathrm{a}}, d_m^{\mathrm{a}})$ measures the disagreement between $c_m^{\mathrm{a}}$ and the corresponding ground truth cell type $d_m^{\mathrm{a}}$. Note that $d_m^{\mathrm{a}}$ also includes the background class, where no ground truth bounding box is available for the (false positive) detected cell, and in this case $J(x_m^{\mathrm{a}}, y_m^{\mathrm{a}})$ is not computed and set to zero.

Since the amount of labeled training data is scarce, the network model trained in the fully supervised setting may not learn adequate knowledge for the detection, and it is possible to improve the network training by incorporating the abundant unlabeled training images. Therefore, we seek to develop a semi-supervised approach that exploits the unlabeled training data to improve the detection performance. An overview of the proposed framework is shown in Fig. 1, and the detailed design of our approach is presented below.

### 2.2. Semi-supervised Cell Detection with Robust Mean Teacher

Motivated by the success of MT (Tarvainen and Valpola, 2017) in SSL, including SSL-based object detection (Liu et al., 2020), we propose to adapt and improve the MT framework for semi-supervised cell detection. Following the MT framework, we construct a teacher model and a student model, which share the same network structure but with different network weights. For convenience, we denote the network weights of the student model and the teacher model by $\theta$ and $\theta'$, respectively.

In the standard MT, the teacher model makes predictions on the unlabeled images, which are considered pseudo-labels for these images, and the student model learns from not only the annotations on the labeled training data but also the pseudo-labels on the unlabeled training data. Both the teacher and student models are updated iteratively during network training. The standard MT is adapted to object detection in (Liu et al., 2020). Specifically, suppose the classification result of the $n$-th detected cell in the unlabeled training images given by the student is $c_n^{\mathrm{u}}$, the corresponding pseudo-label given by the teacher model in the $t$-th iteration is $d_{n,t}^{\mathrm{u}}$, and the total number of the cells detected by the student model

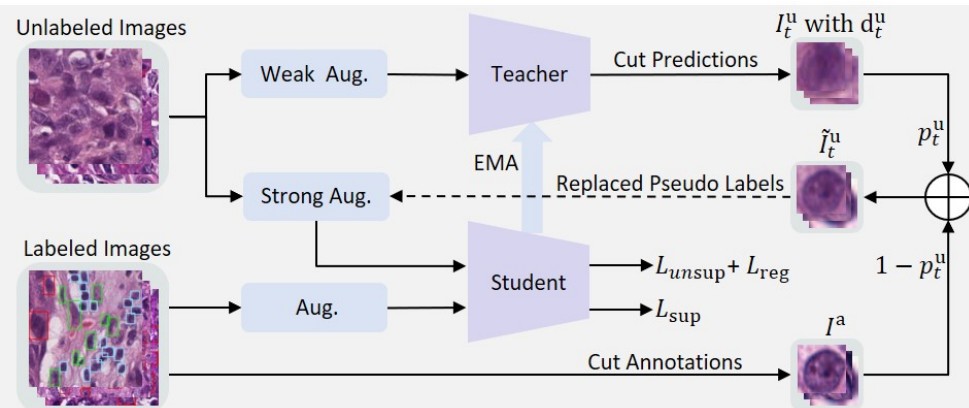

Figure 1: An overview of the proposed framework for semi-supervised cell detection.

in the unlabeled images is $N_{\mathrm{u}}$.[1] Then, in (Liu et al., 2020) an additional unsupervised loss term $\mathcal{L}_{\mathrm{unsup}}^{\mathrm{MT}}$ based on the unlabeled training data is used together with the loss function in Eq. (1) for training the student model at the $t$-th iteration, where

$$\mathcal{L}_{\mathrm{unsup}}^{\mathrm{MT}} = \sum_{n=1}^{N_{\mathrm{u}}} H(c_n^{\mathrm{u}}, d_{n,t}^{\mathrm{u}}). \tag{2}$$

Note that the localization loss is not included as it is observed in (Liu et al., 2020) that the use of the classification loss alone for the unlabeled training images is sufficient.

In the standard MT procedure, as well as its adaptation to object detection in (Liu et al., 2020), the teacher model is initialized with the scarce labeled training data, and thus its predictions on the unlabeled training data can be noisy and inaccurate. This could negatively affect the learning of the student model. Although a high confidence threshold is used in (Liu et al., 2020) to filter out uncertain teacher predictions, it may only preserve a subset of the cells of interest. Therefore, we propose to further improve the MT framework for cell detection so that it is more robust to the label noise in the teacher predictions.

First, we seek to suppress the noise in the teacher prediction. To this end, instead of directly using $c_n^{\mathrm{u}}$ and $d_{n,t}^{\mathrm{u}}$ to compute the unsupervised loss, we propose to mix the unlabeled images with the labeled images to obtain synthetic unlabeled images for network training. For convenience, we denote the image patch corresponding to a cell of interest detected by the teacher model at the $t$-th iteration in the unlabeled images by $I_t^{\mathrm{u}}$, and the image patch corresponding to a randomly selected cell of the same class in the labeled images is represented by $I^{\mathrm{a}}$. Then, for each image patch $I_t^{\mathrm{u}}$, we mix it with the randomly selected $I^{\mathrm{a}}$ as

$$\tilde{I}_t^{\mathrm{u}} = I_t^{\mathrm{u}} \cdot p_t^{\mathrm{u}} + P(I^{\mathrm{a}}) \cdot (1 - p_t^{\mathrm{u}}), \tag{3}$$

where $p_t^{\mathrm{u}}$ is the confidence of the teacher prediction for the image patch $I_t^{\mathrm{u}}$ and $P(\cdot)$ represents the resizing operation with bilinear interpolation to match the patch size of $I^{\mathrm{a}}$ to that of $I_t^{\mathrm{u}}$. The pseudo-label for the synthetic patch $\tilde{I}_t^{\mathrm{u}}$ is still the hard label of the teacher

---

1. Note that weak or strong image perturbation is applied before the teacher or student prediction (Liu et al., 2020), respectively.

prediction. Since there is no label noise in the labeled images, the mixing of labeled patches with unlabeled patches leads to reduced label noise in the synthetic unlabeled patches, and when the teacher prediction on the unlabeled image is less confident, the contribution of the label image is greater in the mixing to suppress the possibly greater label noise.

Each mixed patch $\tilde{I}_t^{\mathrm{u}}$ then replaces the corresponding original image patch $I_t^{\mathrm{u}}$ in the unlabeled images, and the resulting synthetic unlabeled images are used to compute the unsupervised loss. Formally, suppose the classification result of the $n$-th detected cell in the synthetic unlabeled training images given by the student model is $\tilde{c}_n^{\mathrm{u}}$, the corresponding pseudo-label at the $t$-th iteration is $\tilde{d}_{n,t}^{\mathrm{u}}$, and the total number of the cells detected by the student model in the synthetic unlabeled images is $\tilde{N}_{\mathrm{u}}$. Then, we have the following unsupervised loss term at the $t$-th iteration:

$$\mathcal{L}_{\mathrm{unsup}} = \frac{1}{\tilde{N}_{\mathrm{u}}} \sum_{n=1}^{\tilde{N}_{\mathrm{u}}} H(\tilde{c}_n^{\mathrm{u}}, \tilde{d}_{n,t}^{\mathrm{u}}). \tag{4}$$

This loss term can be used in conjunction with $\mathcal{L}_{\mathrm{sup}}$ to train the student model.

In addition to the unsupervised loss term designed above, since there is still noise in the pseudo-labels of the synthetic unlabeled images, we propose to further incorporate a loss term that allows the network training to be more robust to the label noise. In particular, it is theoretically proved in (Zhou et al., 2021) that a loss function can be made robust to noisy labels by restricting the network output to the set of permutations over a fixed vector, and this can be conveniently approximated with the addition of sparse regularization of the output using $\ell_p$-norm ($0 < p \leq 1$). Specifically, this additional term $\mathcal{L}_{\mathrm{reg}}$ is computed for the student predictions on the synthetic unlabeled images as

$$\mathcal{L}_{\mathrm{reg}} = \frac{1}{\tilde{N}_{\mathrm{u}}} \sum_{n=1}^{\tilde{N}_{\mathrm{u}}} \|\tilde{c}_n^{\mathrm{u}}\|_p^p. \tag{5}$$

The complete loss function $\mathcal{L}$ for training the student model at the $t$-th iteration becomes

$$\mathcal{L} = \mathcal{L}_{\mathrm{sup}} + \lambda_{\mathrm{u}} \mathcal{L}_{\mathrm{unsup}} + \lambda_{\mathrm{r}} \mathcal{L}_{\mathrm{reg}}, \tag{6}$$

where $\lambda_{\mathrm{u}}$ and $\lambda_{\mathrm{r}}$ are weights for the loss terms $\mathcal{L}_{\mathrm{unsup}}$ and $\mathcal{L}_{\mathrm{reg}}$, respectively.

After the student model is trained at the $t$-th iteration, like in the standard MT framework, the network weights of the teacher model are also updated based on the student model and the current teacher model with EMA:

$$\theta' \leftarrow \theta' \cdot \sigma + \theta \cdot (1 - \sigma), \tag{7}$$

where $\sigma$ is the EMA decay rate to be specified. The iterative update of the teacher and student models is performed until convergence, and the teacher model is used for the final detection.

Note that our method is different from the previous mixing method MixUp (Zhang et al., 2017) and its adaptation in semi-supervised classification (Berthelot et al., 2019). The previous mixing-based methods are mainly applied in image classification and they stochastically linearly combine two annotated images and the corresponding annotations. However, when MixUp is directly applied to cell detection, the random mixing of two images may lead to undesired interaction between different types of cells or between cells and background. These methods do not help to solve the noise in the teacher model prediction.

### 2.3. Implementation details

We use the Faster R-CNN (Ren et al., 2017) with FPN (Lin et al., 2017) and ResNet50 (He et al., 2016) implemented in Detectron2[2] as the backbone detection network, which is a popular choice for semi-supervised object detection (Sohn et al., 2020; Liu et al., 2020). The weights for feature extraction have been pretrained on ImageNet (Deng et al., 2009) for a better initialization.

Like (Liu et al., 2020), during network training, weak image perturbation is applied to the unlabeled images before the teacher prediction, where random horizontal flipping is used; strong perturbation, including color jittering, Gaussian blur, and random erasing (Zhong et al., 2020; DeVries and Taylor, 2017), is applied to the unlabeled images before the student prediction. *Non-maximum suppression* (NMS) (Neubeck and Van Gool, 2006) is applied for both teacher and student models to remove duplicate predictions. We choose the smooth L1 loss and focal loss to measure the localization error and classification error, respectively (Liu et al., 2020). We set the loss weights $\lambda_u = 4$ and $\lambda_r = 1$ according to (Liu et al., 2020) and (Zhou et al., 2021), respectively; we set $p = 0.5$ for $\mathcal{L}_{reg}$ (Zhou et al., 2021) and the EMA decay rate $\sigma = 0.996$ like in (Liu et al., 2020). The other training configurations, such as the optimizer, learning rate, etc., are set to the default specification in (Liu et al., 2020).

For evaluation, prediction boxes on test images with a confidence score greater than 0.5 are kept and NMS is performed to merge duplicate bounding boxes.

## 3. Results

### 3.1. Dataset Description

To evaluate the proposed method, we performed experiments on the publicly available Nu-CLS dataset (Amgad et al., 2021), which aims to detect multiple types of cells in breast cancer. We used the corrected single-rater subset, which contained 1744 images with annotated cells. The annotation of each cell comprised a bounding box indicating the cell location and the class of the cell. In the original NuCLS dataset, seven types of cells were annotated, and in this work we only selected the three types for which a large number of cells were annotated for evaluation, including the tumor, stromal, and lymphocyte classes.

The images were randomly split into a training, validation, and test set with a ratio of about 7:1:2. The training set was further divided into a labeled training set and unlabeled training set, where the annotation was available and inaccessible during network training, respectively. Specifically, we considered several cases of the labeled training set, where 2%, 5%, 10%, and 20% of the training set was used as the labeled training set and the other training images were used as the unlabeled training set.

### 3.2. Evaluation Results

We compared the proposed method with three competing methods, which, for fair comparison, all used the same backbone Faster R-CNN detection network as the proposed method. The first competing method is the Faster R-CNN model trained with the labeled training set only, where the unlabeled training images were not used. For convenience, this method

---

2. https://github.com/facebookresearch/detectron2

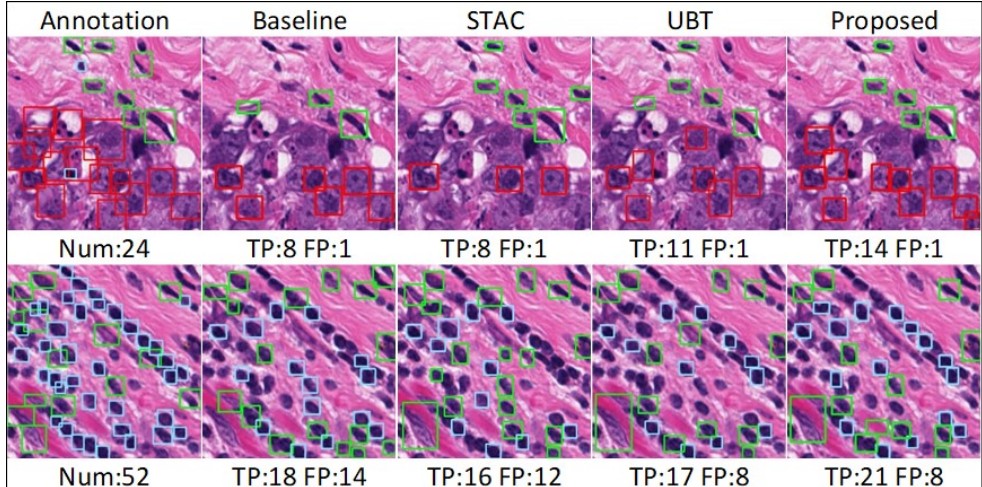

Figure 2: Examples of detection results on test images (achieved with 2% labeled training images) shown together with the annotation. The tumor, stromal, lymphocyte classes are represented by red, green, and blue boxes, respectively. The numbers of *true positive* (TP) and *false positive* (FP) detection results are indicated in the figure for each case. The numbers of annotated cells are also shown for reference.

is referred to as the baseline method. The second competing method is the semi-supervised object detection approach STAC (Sohn et al., 2020) integrated with the baseline Faster R-CNN, where a student model learns from both the labeled and unlabeled training data with a fixed teacher model. The third competing method is the *unbiased teacher* (UBT) method (Liu et al., 2020) developed for semi-supervised object detection based on the mean teacher framework, and it is integrated with Faster R-CNN like the proposed method.

We first qualitatively evaluated the proposed method. Examples of the detection results of each method on test images (achieved with 2% labeled training images) are shown in Fig. 2, together with the annotation for reference. In these cases, our method compares favorably with the competing methods by producing more true positive boxes than the competing methods without increasing the number of false positive boxes.

Next, we quantitatively compared the proposed method with the competing methods. For each case of the training set, we computed the F1-score of the detection results on the test set for each cell type, and the results are shown in Table 1. In all cases, the proposed method has a higher F1-score than the competing methods. In addition, we computed the *mean average precision* (mAP) for the detection results, and the results are shown in Fig. 3. Consistent with Table 1, the proposed method has higher mAP than the competing methods. These results together indicate that the proposed method has better detection accuracy than the competing methods.

Finally, we performed an ablation study to verify the individual benefit of the proposed mixing strategy and the additional sparse regularization. Specifically, we applied our method without the regularization $\mathcal{L}_{\text{reg}}$ in Eq. (5). The mAP for these cases are summarized in Fig. 3 as well. We can see the corresponding mAP is better than the results of

Table 1: The F1-score (%) achieved with different amounts of labeled training data (2%, 5%, 10%, and 20%) for each cell type. The tumor, stromal, lymphocyte classes are represented by Tum, Str, and Lym, respectively. The best results are highlighted in bold.

| Method | 2% | | | 5% | | | 10% | | | 20% | | |
|---|---|---|---|---|---|---|---|---|---|---|---|---|
| | Tum | Str | Lym | Tum | Str | Lym | Tum | Str | Lym | Tum | Str | Lym |
| Baseline | 53.0 | 27.9 | 43.1 | 59.8 | 27.7 | 59.0 | 60.6 | 37.1 | 56.7 | 64.0 | 40.9 | 60.2 |
| STAC | 48.4 | 18.4 | 21.1 | 55.0 | 20.9 | 39.8 | 61.0 | 36.3 | 50.8 | 52.7 | 33.8 | 56.9 |
| UBT | 55.1 | 29.0 | 38.6 | 59.5 | 25.9 | 56.6 | 60.0 | 35.6 | 53.3 | 62.9 | 40.8 | 49.5 |
| Proposed | **56.7** | **31.0** | **53.1** | **60.5** | **35.9** | **61.6** | **62.9** | **41.6** | **61.8** | **65.5** | **43.1** | **62.3** |

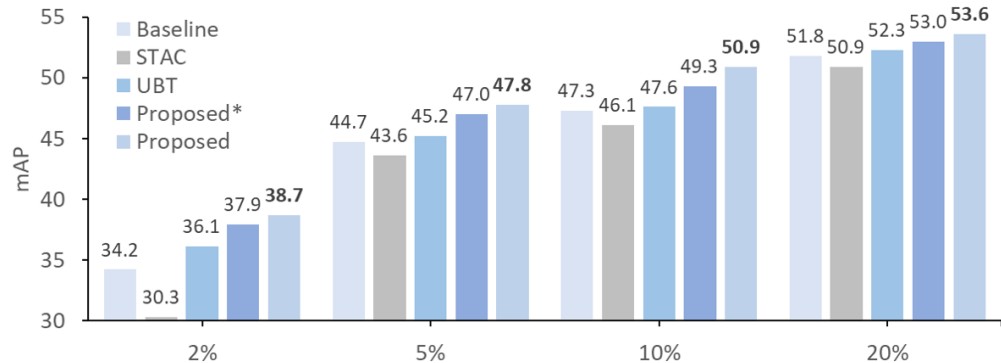

Figure 3: The mAP (%) computed for the detection results for each amount of the labeled training data (2%, 5%, 10%, and 20%). The best results are highlighted in bold. The 'Proposed*' denotes our proposed method without sparse regularization. Our proposed model can efficiently leverage the unlabeled data and perform favorably against the existing semi-supervised object detection works, including STAC and Unbiased Teacher.

the competing methods but worse than the result of the complete proposed method. This result indicates that both of the mixing and sparse regularization in the proposed method are beneficial.

## 4. Conclusion

We have proposed a semi-supervised approach to cell detection in histopathology images. Based on the mean teacher framework, we have a developed a training procedure that is more robust to the noise in the teacher prediction. The experimental results on a publicly available dataset show that our method can improve the performance of semi-supervised cell detection.

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
