# OpenReview forum: "A Robust Mean Teacher Framework for Semi-Supervised Cell Detection in Histopathology Images"
_MIDL.io/2023/Conference — MIDL 2023 Poster_

### Official Review · Reviewer_HiNg · 2023-02-04

**Confidence:** 3
**Preliminary Rating:** 3
**Recommendation:** Poster

**Summary:**

This paper proposes a semi-supervised multi-class cell detection system for histopàthological images based on the Mean-Teacher approach, but attempting to correct for possibly noisy teacher predictions by applying a sort of mix-up mechanism between labeled and unlabeled data. Experiments carried out on public data show some improvement in F1-score with respect to three other techniques.

**Strengths:**

- (Semi-Supervised) Cell detection is one of the computer vision tasks for biomedical image analysis that would benefit strongly from having an automated system, since manually annotating cells is an extremely tedious and error-prone process.
- The idea of regularizing the individual noisy cell detections+classifications on unlabeled data by mixing them up with cells from the same category extracted from annotated data sounds quite interesting and, to my knowledge, original.

**Weaknesses:**

- The main thing I find surprising is the lack of mentioning the parallelisms between the proposed technique (eq. 3, mainly) and the very popular mixup regularization (https://arxiv.org/abs/1710.09412, more than 5,500 citations). Please see below.
- Not being an expert in object detection, I couldn't find any strong weakness, only some lack of information that I would like to ask about. For example, I tried to understand the supervised part of the loss in equation 1, but I fail to see how the model is penalized for missing cells. The loss only loops over detected cells ("N_a=the total number of detected cells in the labeled images" in eq. (1) ) and looks at their distance to correct locations plus their correct categorization. This can also deal with false positives, when a cell is detected but it should have not. However, what happens when a cell is not detected?
- Apparently the proposed method could only deal with three of the original seven cell categories in the dataset, due to struggling with class imbalance, which I guess is a weakness.

**Deanonymize Review:**

no

**Detailed Comments:**

- The idea of adding up an annotated patch and an un-annotated one, giving more weight to confidently predicted patches, eq. (3) sounds good. I understand that the hard label L of the teacher prediction is indeed the label associated to I^a, because using that label L is the way in which you chose I^a from the corresponding category. Therefore you are mixing up the data and keeping the label of the teacher. I would say there is a clear connection with the MixUp regularization technique, and I am quite surprised that this was not mentioned anywhere in the paper. Could you please comment on the similarities between the proposed approach and MixUp?

- Related to this, maybe the authors could try to deal with class imbalance by regularizing using MixUp with different classes from the labeled data and mixing categories, instead of selecting the ones that are the same as the teacher's predictions. MixUp has been found useful for dealing with class imbalance for classification problems in the past (https://doi.org/10.1007/978-3-030-87240-3_31).

- I would advice trying to improve the caption in Figure 1 by expliciting the meaning of the loss functions and the other symbols in the diagram so that the reader does not need to go back to the text looking for the notation.

- I find it quite fascinating that we can use random erasing as a transform here. What happens if we erase a cell? Nothing wrong with that?

**Paper Type:**

methodological development

**Questions To Address In The Rebuttal:**

Mainly I would like to discuss a bit about the relationship with the MixUp technique, and also how the authors deal with False Negatives in their system; I mean, nothing in the proposed loss functions seems to enforce the model to avoid missing cells, I would like to understand how this is possible?

---

### Official Review · Reviewer_MYjH · 2023-02-04

**Confidence:** 3
**Preliminary Rating:** 3
**Recommendation:** Poster

**Summary:**

This paper presents a semi supervised object detection method as applied to cell detection in histopathology images. The main framework is based on the mean-teacher framework from distillation topic. The difference here is that putting some labelled images (changing from unsupervision to semi-supervision) can reduce some of the noise. publicly available data sets were used for evaluations of the presented method. Some promising results were obtained.

**Strengths:**

-- nicely written paper with good smooth reading experience.
-- Mixing the unlabeled images with the label images to produce synthetic unlabeled images for network training is smart idea (but not new) for reducing the effect of noisy teacher.
-- There is a new loss function defined to make the overall system robust to noisy predictions.
-- publicly available data sets used for evaluation, allowing benchmark comparisons with others.

**Weaknesses:**

--  innovation is weak, methods used herein are from already published works.
--  strong and weak augmentation components in Figure 1 require more elaboration.
-- only one data set is used. For generalization evaluation, more data sets will be better fit.


**Deanonymize Review:**

no

**Detailed Comments:**

-- in Table 1, LYM results look too much improvements while others do not. What is the main reason?
-- possible to turn Figure 3 into box plots? bar plots. hide the outliers.
-- can authors demonstrate the effect of chosen labelled images on the final results? (quality vs noise)?
-- how many labeled data is necessary to run the overall system depicted in Figure 1?
-- one data set is weak in terms of generalization.
-- one data set has misleading in generalization. Other data sets would be nice to have for generalization understanding and bias evaluation.


**Paper Type:**

validation/application paper

**Questions To Address In The Rebuttal:**

questions in the detailed comments section, please.
Repeated here:
-- in Table 1, LYM results look too much improvements while others do not. What is the main reason?
-- possible to turn Figure 3 into box plots? bar plots. hide the outliers.
-- can authors demonstrate the effect of chosen labelled images on the final results? (quality vs noise)?
-- how many labeled data is necessary to run the overall system depicted in Figure 1?
-- one data set is weak in terms of generalization.
-- one data set has misleading in generalization. Other data sets would be nice to have for generalization understanding and bias evaluation.

---

### Official Review · Reviewer_MQvN · 2023-02-06

**Confidence:** 4
**Preliminary Rating:** 2

**Summary:**

1. The authors present a semi-supervised object detection technique for detecting cells in histopathology images, built upon and enhancing the mean teacher framework.
2. The standard mean teacher approach can suffer from noisy detection results on unlabeled data from the teacher model, hindering the student model's learning. The paper addresses this by suppressing the noise by combining unlabeled training images with labeled ones with available ground truth detection results.
3. Additionally, the authors suggest incorporating a noise-resistant loss term to enhance the student model's learning from the teacher model.
4. The proposed method was evaluated on a public multi-class cell detection dataset. The results demonstrate that this approach enhances cell detection performance in semi-supervised histopathology image analysis.















**Strengths:**

1. The paper is written clearly and concisely, making it easy to understand. The figures included in the paper effectively illustrate the model and its components, ensuring that the reader understands the content comprehensively.
2. Experiments were conducted on a publicly accessible multi-class cell detection dataset to evaluate the proposed method. The results demonstrate that this technique enhances cell detection performance in semi-supervised histopathology images. The outcomes are encouraging and suggest that the method has the potential for practical applications.
3. The concept of combining unlabeled images with labeled ones to generate synthetic unlabeled images instead of solely relying on predictions or labels appears to be a unique and valuable contribution to the field. The authors' approach to using this novel idea could have significant implications for future research in the area.













**Weaknesses:**

1.  A concern is that there appear to be no ablation studies conducted on the proposed modules, such as the image-mixing operations and the loss terms. This lack of analysis makes it difficult to understand the reasons for the improved results and the extent to which each module contributes to the performance gains. Ablation studies would provide valuable insights into the individual and combined effects of these components and help to build a stronger case for the effectiveness of the proposed method.
2. While the experiments are carried out on a single medium-sized dataset (NuCLS), it would be beneficial to validate the proposed approach on additional datasets or larger ones for greater robustness and reliability. This would help establish the generalizability and practical usefulness of the proposed method for a broader range of applications.





**Deanonymize Review:**

no

**Paper Type:**

methodological development

**Questions To Address In The Rebuttal:**

My initial rating before considering the authors' response is a '2: weak reject'. Although the central concept appears to be novel and the writing is adequate, my main concerns relate to the absence of ablation studies and the limited dataset used for experimentation. I am eager to review the authors' response and would be pleased to upgrade my rating if my concerns are effectively addressed in their reply.

---

### Meta-Review · Area_Chair_wdtG · 2023-02-25

**Recommendation:** Accept (Poster)
**Confidence:** 4

**Metareview:**

This paper addresses cell detection in histopathology images. The authors propose semi-supervised learning based on a mean-teacher approach, attempting to suppress noisy teacher results through a scheme that mixes labeled and unlabeled data. Based on reviewer comments, they have also undertaken more complete ablation experiments and included clarification of the novelty.
Strengths: Reviewers agreed that the paper is clearly written, and that the proposed method represents a meaningful contribution to the field. In the rebuttal, the authors have clarified the differences between their approach and existing methods.
Weaknesses: More data for evaluation would be helpful, though the authors perform systematic experiments using fractions of the dataset used herein. They mention planning to conduct analysis on more data in future work.